# Prediction of odds for emergency cesarean section: A secondary analysis of the CHILD term birth cohort study

**Mon H. Tun**[1], **Radha Chari**[2], **Padma Kaul**[3,4,5], **Fabiana V. Mamede**[1], **Mike Paulden**[4], **Diana L. Lefebvre**[6], **Stuart E. Turvey**[7], **Theo J. Moraes**[8], **Malcolm R. Sears**[6], **Padmaja Subbarao**[8], **Piush J. Mandhane**[1]*

1 Department of Pediatrics, University of Alberta, Edmonton, AB, Canada, 2 Department of Obstetrics and Gynaecology, University of Alberta, Edmonton, AB, Canada, 3 Canadian VIGOUR Centre, University of Alberta, Edmonton, AB, Canada, 4 School of Public Health, University of Alberta, Edmonton, AB, Canada, 5 Department of Medicine, University of Alberta, Edmonton, AB, Canada, 6 Department of Medicine, McMaster University, Hamilton, Ontario, Canada, 7 Department of Pediatrics, University of British Columbia, Vancouver, British Columbia, Canada, 8 Department of Pediatrics, Hospital for Sick Children, University of Toronto, Toronto, Ontario, Canada

* mandhane@ualberta.ca

**Data Availability Statement:** The dataset used for this analysis contains human data of a potentially sensitive nature (e.g. delivery mode, anthropometrics and results from the CES-D). Data

## Abstract

### Introduction

Previously developed cesarean section (CS) and emergency CS prediction tools use antenatal and intrapartum risk factors. We aimed to develop a predictive model for the risk of emergency CS before the onset of labour utilizing antenatal obstetric and non-obstetric factors.

### Methods

We completed a secondary analysis of data collected from the CHILD Cohort Study. The analysis was limited to term (≥37 weeks), singleton pregnant women with cephalic presentation. The sample was divided into a training and validation dataset. The emergency CS prediction model was developed in the training dataset and the performance accuracy was assessed by the area under the receiver operating characteristic curve(AUC) of the receiver operating characteristic analysis (ROC). Our final model was subsequently evaluated in the validation dataset.

### Results

The participant sample consisted of 2,836 pregnant women. Mean age of participants was 32 years, mean BMI of 25.4 kg/m2 and 39% were nulliparous. 14% had emergency CS delivery. Each year of increasing maternal age increased the odds of emergency CS by 6% (adjusted Odds Ratio (aOR 1.06,1.02–1.08). Likewise, there was a 4% increase odds of emergency CS for each unit increase in BMI (aOR 1.04,1.02–1.06). In contrast, increase in maternal height has a negative association with emergency CS. The final emergency CS delivery predictive model included six variables (hypertensive disorders of pregnancy,

used for this analysis are available through an application to the CHILD study (https://childstudy. ca/for-researchers/data-access/). Researchers interested in accessing our data should refer to CHILDdb: C104 "Prediction of mode of delivery in the CHILD birth cohort" when requesting the data.

**Funding:** This study was supported by the Canadian Institutes of Health Research (CIHR), the Women's and Children's Health Research Institute (WCHRI) and the Allergy Genes and Environment Network Centres of Excellence (AllerGen NCE). The funders had no role in study design, data collection and analysis, decision to publish, or preparation of the manuscript.

**Competing interests:** The authors report no conflict of interest.

antenatal depression, previous vaginal delivery, age, height, BMI). The AUC for our final prediction model was 0.74 (0.72–0.77) in the training set with a similar AUC in the validation dataset (0.77; 0.71–0.82).

## Conclusion

The developed and validated emergency CS delivery prediction model can be used in counselling prospective parents around their CS risk and healthcare resource planning. Further validation of the tool is suggested.

## Introduction

The World Health Organization (WHO) has raised concerns regarding the dramatic increase in cesarean section (CS) rates. CS is effective in managing dystocia and other significant complications of pregnancy. Indications for scheduled CS can be divided into absolute and relative indications [1]. Absolute indications for scheduled CS include cephalopelvic disproportion, placenta previa, abnormal lie and presentation. Prior CS delivery is a relative indication for scheduled CS [1–5].

Emergency CS is indicated when acute obstetrical complications that threaten the life of the mother and/ or the fetus including fetal distress and antepartum hemorrhage develop. Intrapartum factors such as labour dystocia, fetal distress, and umbilical cord prolapse are absolute indications for emergency CS [1, 6, 7]. Emergency CS, is associated with increased maternal morbidity and mortality, compared to a scheduled CS. Morbidity associated with emergency CS include severe hemorrhage, complications from rapid administration of general anesthesia and accidental injury to the mother and infant [8–11]. A meta-analysis reported that the rates of maternal and fetal complications and mortality were higher in emergency CS when compared to scheduled CS [12]. In addition to the additional morbidity and mortality, resource planning for an emergency CS is more difficult compared to scheduled CS resulting in higher infection rates [9].

The CS risk prediction model developed by Janssen et al and Souza et al utilized both antenatal and intrapartum factors for low risk nulliparous pregnant women [13, 14]. The FLAMM scoring system, developed to predict a VBAC (vaginal birth after prior cesarean section), included intrapartum factors including cervical dilation and effacement. The Grobman calculator [15], which included only antenatal factors, has limited generalizability as the tool is meant to predict the probability of a vaginal birth after cesarean section (VBAC) for term pregnant women with one prior CS. Tools to predict emergency CS delivery have incorporated antepartum and intrapartum factors [16, 17]. The emergency CS risk prediction model and classification tree (CTREE), with discriminatory accuracy ranges from 0.74 to 0.81, included intrapartum factors such as scalp pH, and labour induction among women with history of previous CS [18]. A risk scoring system for emergency CS was developed utilizing both antenatal and intrapartum factors such as quantity and characteristic of the amniotic fluid in Chinese population [3]. We could not identify a tool or scoring system for emergency CS risk prediction utilizing prenatal factors only. In this study, we used data from the CHILD Cohort Study to identify the main antenatal obstetric and non-obstetric risk factors for emergency CS and to subsequently develop an emergency CS prediction tool.

## Materials and methods

### Study design and participants

This was a secondary analysis of the CHILD Cohort Study, a large general-population recruited prospective observational pre-birth cohort study of 3,455 pregnant women enrolled in Edmonton, Winnipeg, rural Manitoba, Vancouver and Toronto between 2009 and 2012. This secondary analysis focused primarily on the emergency CS prediction. Details on the data collection methods and the characteristics of the cohort have been described previously [19] (www.childstudy.ca). Mothers were approached for enrollment in the study during the second or third trimester of their pregnancy. Infants, and their parents, were recruited if born at 34 weeks' gestation or later and with birth weight of 2,500 g or more.

Mothers completed questionnaires on general health such as diabetes, hypertension and psychosocial factors at the time of recruitment and at 36 weeks of gestation. Information regarding maternal age, weight (kg), height (cm), parity, socioeconomic status, maternal education, ethnicity, maternal smoking status, medical comorbidities and risk factors including hypertensive disorder [20] and diabetes mellitus complicating pregnancy [21] were collected through standardized questionnaire. Maternal antenatal depression was assessed using the Center for Epidemiologic Studies Depression Scale (CES-D) [22]. Participants were classified as depressed if their CESD-score was ≥10 points. The socioeconomic status (SES) was divided into two groups with a cut-off income of ≥ $60,000 which indicates higher socioeconomic status. Maternal early pregnancy BMI was calculated using self-reported height and weight and classified by World Health Organization (WHO) criteria. Delivery information, including delivery mode, gestational age at birth and neonatal sex, were obtained from birth chart reviews. Pregnant women provided written informed consent to participate in the CHILD study. Ethics approval was obtained from the research ethics board of each CHILD study center (University of Alberta research ethics board, McMaster University research ethics board, Hospital for Sick Children research ethics board, University of Manitoba research ethics board, and University of British Columbia research ethics board) in addition to the McMaster University research ethics board. Patients provided informed written consent to have data from their medical records used for the CHILD study. A separate ethics approval was obtained from the University of Alberta Research Ethics Board for this secondary data analysis (Pro00092920). The data used for this study was de-identified (no participant were identifiers included in the dataset) prior to being released for analysis.

The present study restricted the analysis to nulliparous and multiparous women carrying a singleton, cephalic presentation fetus at 37 completed weeks of gestation with available birth chart records (n = 3,408). The remaining exclusion criteria were: women with a higher risk of scheduled CS such as placenta previa, a prior CS delivery, multiple gestation, cephalopelvic disproportion, breech presentation, pre-term, home birth and those who had their labour induced.

### Statistical analysis

Data analysis steps to develop an emergency CS score are described in **Fig 1**. A total of 2,836 pregnant women met the inclusion criteria. First, the data were randomly divided into two groups: a training dataset (80% of the sample, n = 2,269) and a validation dataset (20% of the sample, n = 567). The demographic, antenatal and obstetric characteristics of training and validation data set was shown in **S1 Table in S1 File**. Categorical variables were analyzed using the Chi-squared test or the Fisher's exact test and t-test was employed for the continuous variables. All parametric data were expressed as mean ± standard deviation (SD), and non-parametric

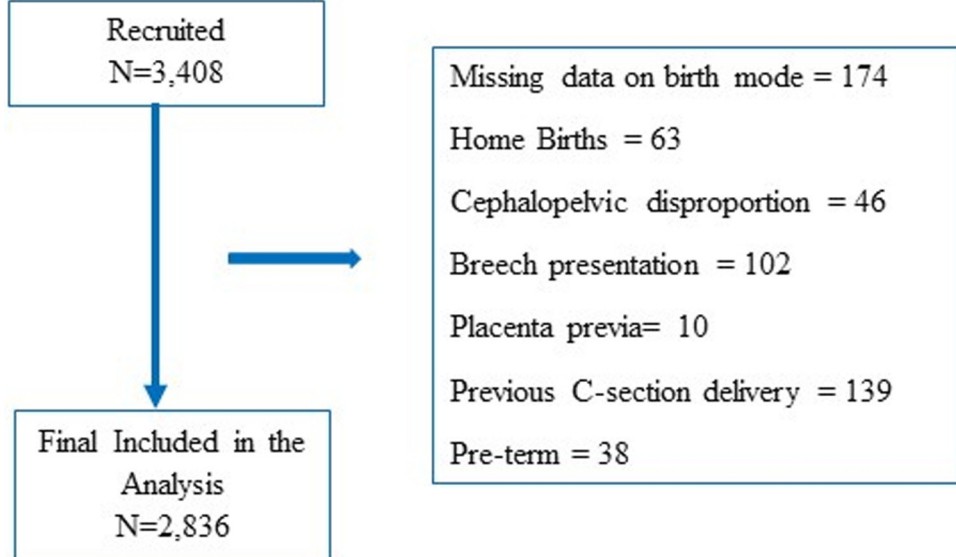

**Fig 1. Flow diagram of the selection of study cohort included in the prediction model.**

data as median ± interquartile range (IQR). The primary outcome was emergency CS for any indication. Mean centering was employed to center the maternal age and height variables.

In the training dataset, the univariate and multiple logistic regression models were used to determine the factors associated with emergency CS. The predictors considered for the model included maternal age, ethnicity, height, weight, BMI, gestational age at delivery and parity. Variable selection for the CS risk prediction was based on combination of literature review **(S2 Table and S1 Appendix in S1 File)**, clinical experience and found to be significant in univariate analysis. The variables with a p-value of <0.20 in univariate analyses were included in the multiple logistic regression model. A prediction model (vaginal vs. emergency CS) was then developed with the training data set taking hospital or province difference of cesarean section rate into account. The C-statistic, area under Receiver operating characteristic curve (AUC), was used to assess the performance of the prediction model based on the model's sensitivity and specificity. The final model was adjusted for maternal height, BMI, CESD-score, hypertensive disorders of pregnancy, history of previous vaginal delivery and hospital CS rate.

The predictive ability of the model was then evaluated in the validation data set. The *p*-values for all hypothesis tests were 2-sided and statistical significance was set at *p* <0.05 for all analyses. Goodness-of-fit for the logistic regression models was assessed by using the Hosmer and Lemeshow test. The scoring system was developed based on the weighted estimate of the multiple logistic regression model. The flow diagram of the statistical analysis was described in **S1 Fig in S1 File**. Data analysis was carried out using STATA version 14.

## Results

The demographic and clinical characteristic of women in the study cohort are presented in **Table 1**. Of the 2,836 low-risk pregnant women included in the final analysis, 22% had a CS delivery with 14% (365/2680) emergency CS delivery. The majority of women enrolled were Caucasians (73%). The mean age of women at enrollment was 32 years with a mean BMI of 25 kg/m$^2$. Among infants delivered by emergency CS, 59% (214/365) were male. Among the women delivered by emergency CS, 6% had gestational diabetes, 7% had hypertensive

**Table 1. Demographic, antenatal and obstetric characteristics associated with mode of delivery.**

| Characteristics | Vaginal * (n = 2,315) | Emergency CS (n = 365) | Scheduled CS (n = 156) |
|---|---|---|---|
| Maternal Age (years) (mean ± SD) | 31.99 ± 4.62 | **32.63 ± 4.86** | **33.70 ± 4.30** |
| Maternal Height (cm) (mean ± SD) | 165.53 ± 6.81 | **162.79 ± 6.98** | 164.60 ± 7.50 |
| Maternal Weight (kg) (mean ± SD) | 68.75 ± 16.42 | **70.90 ± 18.37** | **73.06 ± 20.36** |
| BMI in kg/m² (mean ± SD) | 25.07 ± 5.68 | **26.71 ± 6.46** | **26.89 ± 6.89** |
| Hospitals CS rate CHILD cohort (mean ± SD) | 5.93 ± 3.20 | **6.57 ± 3.16** | 7.4 ± 3.30 |
| Increased CESD-score (Ref: <10) | 614 (27%) | **128 (35%)** | 49 (31%) |
| Gestational Age (weeks) | | | |
| 37 | 133 (6%) | **29 (8%)** | **9 (6%)** |
| 38 | 253 (11%) | **42 (12%)** | **35 (23%)** |
| 39 | 551 (24%) | **65 (18%)** | **75 (49%)** |
| 40 | 754 (33%) | **92 (26%)** | **29 (18%)** |
| 41 | 514 (22%) | **104 (29%)** | **5 (2.5%)** |
| ≥42 | 97 (4%) | **27 (7%)** | **1 (0.5%)** |
| Gravida | | | |
| G1 | 862 (37%) | **199 (55%)** | **38 (24%)** |
| G2 | 748 (32%) | **90 (25%)** | **56 (36%)** |
| G3 | 386 (17%) | **38 (10%)** | **34 (22%)** |
| G4 | 176 (8%) | **21 (6%)** | **14 (9%)** |
| ≥G5 | 142 (6%) | **16 (4%)** | **14 (9%)** |
| Maternal Ethnicity | | | |
| Caucasian | 1334 (74%) | **157 (67%)** | 111 (74%) |
| Others | 462 (26%) | **77 (33%)** | 40 (26%) |
| Marital status | | | |
| Married or Common Law | 1982 (86%) | 302 (83%) | 136 (87%) |
| Single (Never been married) | 113 (5%) | 7 (5%) | 17 (5%) |
| Divorced/Widowed/ Separated | 220 (9%) | 56 (12%) | 3 (8%) |
| Socioeconomic status | | | |
| <$60,000 | 418 (21%) | 62 (19%) | **18 (14%)** |
| ≥ $60,000 | 1592 (79%) | 258 (81%) | **115 (86%)** |
| Maternal Education | | | |
| No education beyond high school | 209 (9%) | 30 (9%) | **6 (4%)** |
| Some post secondary/ college | 448 (20%) | 87 (25%) | **41 (28%)** |
| University degree | 1559 (71%) | 235 (67%) | **98 (68%)** |
| Maternal smoking history | | | |
| Yes | 133 (7%) | 19 (8%) | 14 (9%) |
| Hypertensive Disorders of Pregnancy | | | |
| Yes | 69 (3%) | **26 (7%)** | 6 (4%) |
| Gestation Diabetes | | | |
| Yes | 98 (4%) | 23 (6%) | 9 (6%) |
| Previous vaginal delivery | | | |
| First Born | 1170 (51%) | **289 (79%)** | **47 (30%)** |
| Subsequent Born | 1141 (49%) | **75 (21%)** | **109 (70%)** |
| Child Sex | | | |
| Male | 1204 (52%) | **214 (59%)** | 79 (51%) |
| Female | 1111 (48%) | **151 (41%)** | 77 (49%) |
| Analgesia | | | |
| Epidural | 1414 (61%) | **275 (75%)** | **10 (6%)** |

(*Continued*)

**Table 1.** (Continued)

| Characteristics | Vaginal * (n = 2,315) | Emergency CS (n = 365) | Scheduled CS (n = 156) |
|---|---|---|---|
| Spinal | 27 (1%) | **90 (25%)** | **151 (97%)** |
| General Anesthesia | 5 (0.2%) | **23 (6.3%)** | **5 (3%)** |

* = Vaginal delivery was used as a reference and compared with emergency CS and scheduled CS.

**p-values** <**0.05 in bold**; SD = standard deviation; BMI = body mass index, PE = preeclampsia.

disorders of pregnancy and 20% of the babies were delivered before reaching full-term ($\geq$ 39 weeks). Women delivered by emergency CS had greater depression symptoms (CESD-scores $\geq$ 10 points) than the vaginally delivered group (35% vs. 27%, p = 0.0001). The women with emergency CS were older and had higher BMI when compared to vaginally delivered women (33.70 vs. 31.99, p = 0.01; 32.63 vs. 31.99, p = 0.023). **S1 Table in S1 File** shows the clinical characteristics of considered predictors in the training and validation data set.

In multiple logistic regression, women with antenatal depression score $\geq$ 10 points had a 45% increased risk of being delivered by emergency CS (aOR 1.45, 1.07–1.96). Each year of increasing maternal age increased the odds of CS by 6% (aOR 1.06, 1.02–1.08) and each unit increase in BMI increased the odds of CS by 4% (aOR 1.04, 1.02–1.06). Pregnant women who had an emergency CS were more likely to have hypertensive disorders of pregnancy (aOR 1.75, 1.01–3.07). In contrast, taller pregnant women (aOR 0.94, 0.92–0.96) and women who had a previous vaginal delivery had lower odds of having an emergency CS (aOR 0.21, 0.15–0.30). Women who had history of previous vaginal delivery (aOR 0.46, 0.36–0.59) was significantly associated with decreased risk of CS (**Table 2**). In our stratified analysis by parity, hypertensive disorders of pregnancy was a significant predictor for CS in nulliparous but not multiparous mwomen (aOR 1.93, 1.02–3.67 vs. aOR 1.09, 0.55–2.21) (**S3 Table in S1 File**). We also conducted sensitivity analysis with the exclusion of CESD-score variable from our prediction model (**S4 Table in S1 File**) and the findings are comparable to the full model.

**Table 2. Multiple logistic regression results include demographic, antenatal physical and obstetric characteristics in overall cohort independent of the parity: (Training and validation dataset).**

| | Emergency CS (Training, n = 2150) | | | Emergency CS (Validation, n = 530) | | |
|---|---|---|---|---|---|---|
| | Coefficient | Odds Ratio | 95% CI | Coefficient | Odds Ratio | 95% CI |
| Centered Age (years) | **0.05** | **1.06** | **1.02–1.08** | **0.13** | **1.14** | **1.07–1.22** |
| Centered Height (cm) | **-0.06** | **0.94** | **0.92–0.96** | **-0.06** | **0.94** | **0.90–0.98** |
| BMI in kg/m$^2$ | **0.04** | **1.04** | **1.02–1.06** | **0.07** | **1.07** | **1.03–1.12** |
| CESD-score (ref: <10) | **0.37** | **1.45** | **1.07–1.96** | **0.51** | **1.66** | **1.01–3.15** |
| Hospital CS rate (CHILD) | 0.04 | 1.04 | 0.98–1.09 | 0.16 | 1.17 | 0.98–1.28 |
| Hypertensive Disorders of Pregnancy | **0.56** | **1.75** | **1.99–3.08** | 0.28 | 1.32 | 0.36–4.80 |
| Previous vaginal delivery | **-1.57** | **0.21** | **0.15–0.29** | **-1.63** | **0.20** | **0.10–0.38** |
| Constant | -2.85 | | | -4.33 | | |
| AUC | | 0.74 | 0.72–0.77 | | 0.77 | 0.71–0.82 |
| Sensitivity | | 12% | | | 13% | |
| Specificity | | 99% | | | 98% | |
| Positive Predictive Value | | 28% | | | 63% | |
| Negative Predictive Value | | 87% | | | 89% | |
| Accuracy | | 87% | | | 85% | |

**P-values** <**0.05 in bold;** AUC = area under curve; OR = odds ratio; CI = confidence interval

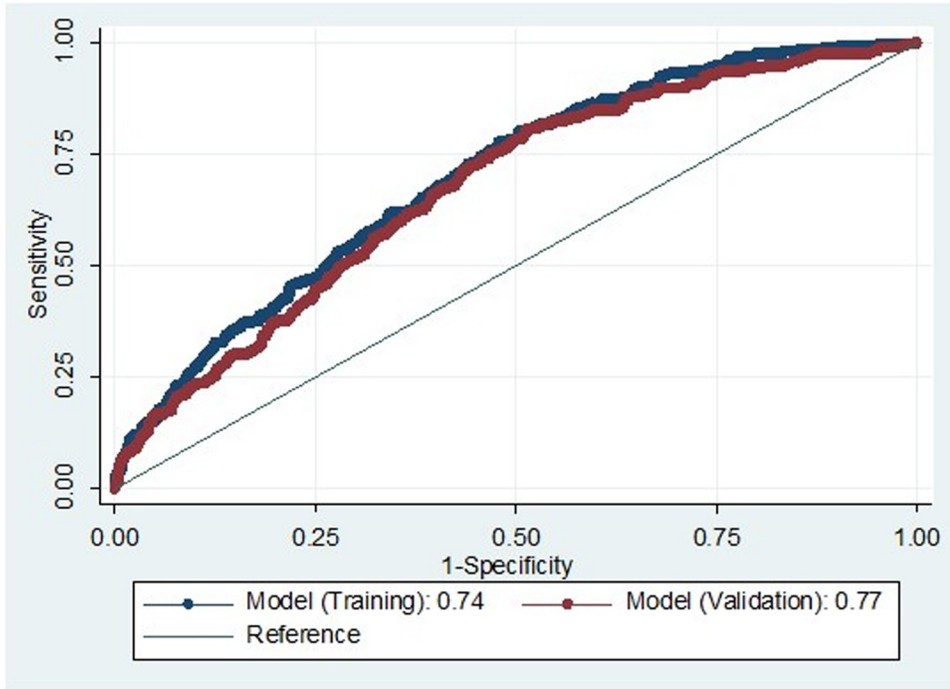

**Fig 2. Comparison of the ROC curve for internal validation (training vs. validation) from multiple logistic regression: Emergency CS.**

Our emergency CS model identified six predictors when controlling for hospital delivered: maternal age, height, BMI, hypertensive disorders of pregnancy, antenatal depression score (CES-D), previous history of vaginal delivery (**Table 2**). The AUC values for the development prediction models was 0.74 (0.72–0.77) while the AUC for the validation dataset was 0.77 (0.71–0.82) (**Table 2**, **Fig 2**). The calibration curve of the prediction model was presented in **Fig 3**. We subsequently developed a modified scoring system based on the logistic regression model coefficients that ranged from 0 to 14 (**Table 3**). The scores were further categorized into grade 0 (0–5 points), grade 1 (6–7 points), grade 2 (8–9 points), and grade 3 ($\geq$ 10 points). With the increase in grade, there was an increase in odds of emergency CS risk (**Table 4**). For example, individuals with grade 2 had a 6.11 increased odds of having an emergency CS (95% CI; 3.06–12.19) while individuals with grade 3 had a 13.96 increased odds of an emergency CS (95%CI; 7.32–26.61) compared to individuals with grade 0 (baseline) risk. The developed modified scoring system provided a sensitivity of 11%, specificity of 91% and an AUC of 0.70 (0.68–0.73) (**Table 4**). Among women with a grade 1 risk of an emergency CS, the number needed to treat (NNT) is seven (i.e. schedule seven CS to prevent one emergency CS), while the NNT was three for emergency CS grade 2 while NNT = 4 and women with a grade 3 emergency CS risk. We also developed an emergency CS risk prediction calculator in the Redcap which can be utilized by the healthcare professionals to assess the risk and provide counselling to the high risk pregnant women.

## Discussion

We developed a score that identifies low-risk pregnant women at risk for an emergency cesarean using data from a large population based cohort from different sites in Canada. The score includes antenatal obstetric and non-obstetric factors, as well as birth order of the infant, and

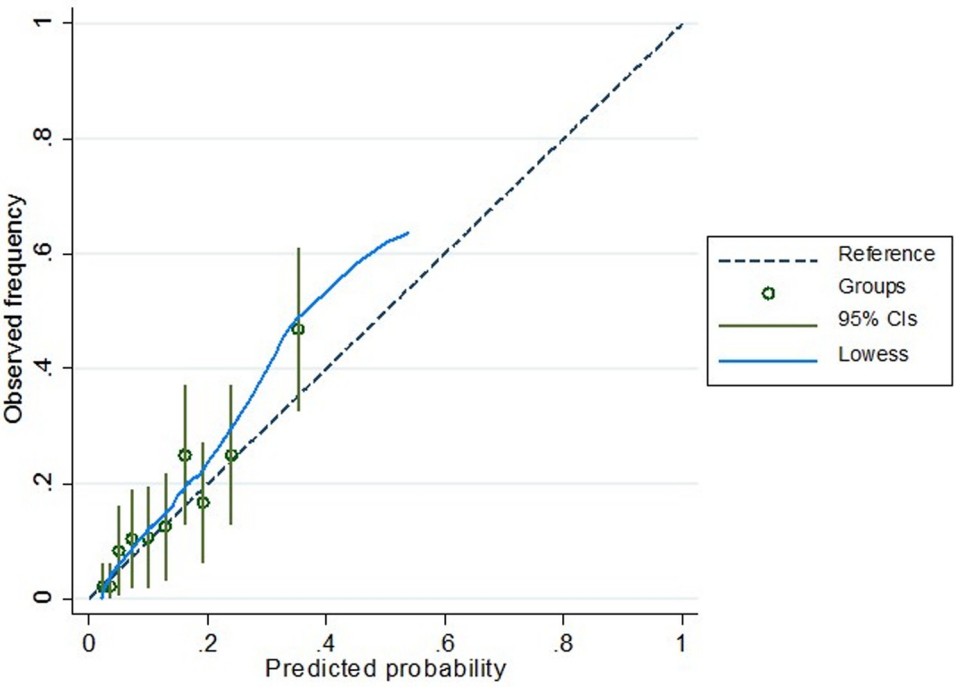

**Fig 3. Calibration curve from multiple logistic regression: Emergency CS.**

controls for the hospital CS rate. The yielded AUC are comparable to prediction models that included intrapartum factors [14, 23], birth weight of the infants [23] and premature rupture of membrane [23]. Most of the parameters in our predictive model are routinely collected as part of routine prenatal care except the CES-D (maternal depression) score. Furthermore, our model has good generalizability as the score was developed from deliveries from 13 different hospitals distributed across Canada. The emergency CS scores could be utilized in the overall context of clinical information to help patient with counseling, expectation and decision-making.

Several studies, including our own, have shown that advanced maternal age was associated with higher odds of having a CS delivery [14, 23–25]. Similarly, our finding of an inverse association between maternal height and CS delivery is consistent with prior studies [12, 21, 24, 26]. Furthermore, a higher maternal BMI has been associated with adverse obstetric outcome and increased the risk of CS delivery [27, 28]. A previous history of vaginal delivery decreased the risk of emergency CS were consistent with the findings from VBAC prediction models [24, 29]. In contrast to prior studies, we did not find that sociodemographic factors such as ethnicity, education and social class and employment and income status were associated with emergency CS [17, 30]. The CS prediction models from Souza et al, a multicenter study, included

**Table 3. Modified antenatal scoring system for predicting the risk of Emergency CS.**

| Age (years) | | Height (cm) | | BMI (kg/m2) | | CES-D score | | Previous vaginal delivery | | Hypertensive Disorders of Pregnancy | |
|---|---|---|---|---|---|---|---|---|---|---|---|
| Value | Score | Value | Score | Value | Score | Value | Score | Value | Score | Value | Score |
| $\leq 30$ | 0 | $\leq 160$ | 4 | $< 18.5$ | 0 | Low (<10) | 0 | Absent | 5 | Absent | 0 |
| 31–35 | 2 | 161–165 | 2 | 18.5–25 | 1 | High ($\geq$10) | 2 | Present | 0 | Present | 2 |
| > 35 | 4 | > 165 | 0 | > 25 | 3 | | | | | | |

**Table 4. Emergency CS prediction risk scoring system.**

| Score | n (%) | Emergency CS (n, %) | Odds Ratio (95% CI) | Numbers Needed to Treat (NNT) |
|---|---|---|---|---|
| Grade 0 (0–5 points) | 459 (22%) | 10 (3%) | Reference | - |
| Grade 1 (6–7 points) | 353 (16%) | 24 (8%) | 3.28 (1.55–6.94) | 7 |
| Grade 2 (8–9 points) | 434 (20%) | 52 (18%) | 6.11 (3.06–12.19) | 3 |
| Grade 3 (≥10 points) | 898 (42%) | 213 (71%) | 13.96 (7.32–26.61) | 4 |

AUC: 0.70 (0.68–0.73)

both maternal and fetal antenatal and intrapartum factors such as cervical position, fetal station, fetal distress and fetal head molding [13]. The AUC of our prediction model containing only antenatal factors (074, training, 0.77 validation) was comparable to the model developed by Souza et al (AUC 0.78). The emergency CS risk scoring system by Guan et al (2020) [3], with an AUC of 0.79, included intrapartum factors such as quantity and color of the amniotic fluid.

Similar to the findings from the previous studies [31–34], we found that multiparous with a history of previous vaginal delivery and with cephalic presentation had lower risk of CS delivery. Hypertensive disorders of pregnancy increased the risk of CS delivery [35]. Nulliparous women have significant higher risk of developing hypertensive related disorders than multiparous women [36, 37]. Similarly, we found pregnancy-induced hypertension increased the risk of emergency CS among nulliparous but not multiparous pregnant women. Nonetheless, unforeseen circumstances such as prolonged labour and fetal distress can occur in multiparous women with prior vaginal delivery. Emergency CS is indicated when acute complications like fetal distress and antepartum hemorrhage develop and threaten the life of the mother and/ or the fetus. Careful assessment and monitoring during antenatal and intranatal period should be provided to both nulliparous and multiparous to improve maternal and neonatal outcome. Our prediction tool specifically excluded intra-partum factors associated with emergency CS such as labour dystocia and fetal distress. The contribution of labour dystocia and fetal distress on emergency CS rates are increasing area of focus for intervention. The American College of Obstetricians and Gynecologists (ACOG) guidelines for reducing the primary CS due to labour dystocia [1] were developed in 2014. As such, these guidelines were not implemented in the Society of Obstetricians and Gynaecologists of Canada (SOGC) guidelines [38] during the study recruitment period (2009–2012) [1]. Further work will examine the role of our tool in the context SOGC guidelines for management of labour dystocia and fetal distress.

The components of our eCS score including maternal age, height, BMI, childbirth history and hypertensive disorders of pregnancy are often collected as part of routine prenatal care. Additionally, unlike prior emergency CS tools, our score does not include any intrapartum data allowing for application at any point during pregnancy. The antenatal obstetric and non-obstetric factors identified from our prediction tool can be utilized in screening and identification of individuals at high risk for an emergency CS. Increase surveillance and antennal interventions could be provided for the modifiable antenatal risk factors such as low dose aspirin for hypertension, counseling for depression and weight management for overweight pregnant women.

Unique to our study, we observed that women with higher antenatal depression score had higher risk of emergency CS delivery. One study reported that mental health status, in particular stress, sleep disturbances and worry were associated with higher risk of emergency CS [39]. Fear and anxiety of childbirth [40, 41] and depressed mood [42] are common causes for preference for CS. Our study finding suggests clinicians should assess for the presence of antenatal

depression in routine antenatal screening for emergency CS risk. In addition, comprehensive mental health programs and the effective interventions of health promotion could reduce the fear and promote confidence with childbirth by vaginal delivery. We were not able to develop a scoring system for scheduled CS with a significant predictive capacity. The Avon Longitudinal Study reported that the largest impact on scheduled CS was breech presentation and previous CS [17]. Our exclusion of women with breech presentation, prior CS delivery, placenta previa and cephalopelvic disproportion, and abnormal lie and presentation from the analysis, known risk factors for a scheduled CS [4], may have resulted in the inability of our scoring system to predict the risk of scheduled CS. Additionally, the data on prior CS history is incomplete in our study population. Hence, we cannot be certain whether the observed increased in the risk of scheduled CS in the subsequent born children could be a confounding effect of prior CS history.

Strengths of our study include a nationwide, prospective design, conducted in a large birth cohort study from four sites in Canada. With the multinomial logistic regression model, the risk of emergency CS were estimated and the parameter estimates are more efficient with less error. Our study had access to the wide range of sociodemographic and pregnancy related variables beyond what would normally be available in a clinical chart review. In addition, many of the antenatal factors utilized for the prediction model were verified with birth chart review by research assistant. Finally, the large sample size provided us with sufficient power to predict emergency CS risk and develop the scoring system with internal validation.

Our research is not without limitation. The observational study design with self-reported items may introduce systematic error in the variance of the predictor variables. Our study did not have access to complete information on maternal weight change during pregnancy, presence of oligohydramnios and estimated fetal weight. We only included term infants in this analysis as the risk factors for CS are different in pre-term infants. Our prediction model included both nulliparous and multiparous pregnant women which may have impact on model. Nonetheless, we adjusted for birth order in our prediction model as well as undertaking sensitivity analysis in nulliparous and multiparous subgroups. We did not find that socioeconomic status impacted our findings. This may be the result of a higher socioeconomic status study sample or a reflection of our publicly-funded healthcare system. Our observed lower CS rate may be due to the lower proportion of overweight women in the study. Lack of information on estimated fetal weight during the third trimester may limit the prediction ability of our model.

While we performed internal validation by splitting the data set, we lacked data for conducting external validity for our CS prediction model. Future research could include external validation of the score in other large, prospectively cohort study. The lack of complete information on prior CS will be worth exploring as an explanation of the variation in scheduled CS and the role of women's preferences. Subsequent work may assess the impact of our prediction model in decision-making about timing and mode of delivery and thereby influence acute and long-term outcomes for women and their offspring. The CESD-score used in the prediction model are not routinely collected during antenatal care. Further studies may consider validating our emergency CS prediction tool with routinely collected antenatal depression questions.

The high specificity and low sensitivity suggest that the tool is good at determining who will not need an eCS (a rule-out test). As such, we would recommend that women who screen positive have closer pre-natal follow-up. The tool identifies areas for potential intervention to lower an individual's risk for an eCS. Our study indicated that women with a higher BMI were more likely to have an emergency CS delivery and. Future research will examine weight control efforts before and during pregnancy may help to reduce the emergency CS rate [27].

## Conclusions

We successfully developed a model to predict the likelihood of emergency CS using prenatal obstetric and non-obstetric factors. The proposed prediction model has similar performance characteristics compared to other emergency CS prediction models without the need for intrapartum prediction factors. The tool could be used in conjunction with the Grobman VBAC tool [15] to assist in delivery mode decision-making and healthcare resource planning and allocation. Early identification of the women at an increased risk of emergency CS is important for patient management including referral for mental health counseling and weight management program to prevent emergency CS. Further prospective validation studies in the general population should be undertaken to confirm efficacy of the developed prediction model and the scoring system before application in the general population.

## Supporting information

**S1 File.**
(DOCX)

## Acknowledgments

We are grateful to all the families who took part in this study, and the whole CHILD team, which includes interviewers, data and laboratory technicians, clerical workers, research scientists, research assistants, volunteers, managers, receptionists and nurses.

## Author Contributions

**Conceptualization:** Mon H. Tun.

**Formal analysis:** Mon H. Tun.

**Funding acquisition:** Piush J. Mandhane.

**Methodology:** Mon H. Tun.

**Supervision:** Piush J. Mandhane.

**Validation:** Mon H. Tun.

**Visualization:** Mon H. Tun.

**Writing – original draft:** Mon H. Tun.

**Writing – review & editing:** Mon H. Tun, Radha Chari, Padma Kaul, Fabiana V. Mamede, Mike Paulden, Diana L. Lefebvre, Stuart E. Turvey, Theo J. Moraes, Malcolm R. Sears, Padmaja Subbarao, Piush J. Mandhane.

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
