## [Decision Letter · Decision Letter 0]

1 Jul 2021

PONE-D-21-10805

Prediction of odds for emergency cesarean section: a secondary analysis of the CHILD term birth cohort study

PLOS ONE

Dear Dr. Mandhane,

Thank you for submitting your manuscript to PLOS ONE. After careful consideration, we feel that it has merit but does not fully meet PLOS ONE’s publication criteria as it currently stands. Therefore, we invite you to submit a revised version of the manuscript that addresses the points raised during the review process.

We look forward to receiving your revised manuscript.

Kind regards,

Eduardo Ortiz-Panozo, MD; MSc

Academic Editor

PLOS ONE

Journal Requirements:

2. Thank you for including your ethics statement: "Ethics approval was obtained from local authorized review board of each CHILD study center and McMaster University. A separate ethics approval was obtained for this secondary data analysis (Pro00092920)." 

c) Please provide additional details regarding participant consent. In the ethics statement in the Methods and online submission information, please ensure that you have specified (1) whether consent was informed and (2) what type you obtained (for instance, written or verbal, and if verbal, how it was documented and witnessed). If your study included minors, state whether you obtained consent from parents or guardians. If the need for consent was waived by the ethics committee, please include this information.

3. In ethics statement in the manuscript and in the online submission form, please provide additional information about the patient records/samples used in your study. Specifically, please ensure that you have discussed whether all data/samples were fully anonymized before you accessed them and/or whether the IRB or ethics committee waived the requirement for informed consent. If patients provided informed written consent to have data/samples from their medical records used in research, please include this information.

Additional Editor Comments:

Since the study aims at developing a prediction model, I suggest to include a table showing the final model's regression coefficients plus the intercept. Not as a supplementary table (i.e, Supplementary table 3) but as part of the main analysis.

Please clarify how many and what variables were tested as potential predictors. What were the p-values for excluded variables?

Authors mention Table 3a and a scoring system in the text, but I could not find those results in tables.

I would suggest to include more statistical measures of the predictive capacity of the model. Is it possible to calculate diagnostic accuracy, sensitivity, specificity, PPV, NPV and/or diagnostic likelihood ratios?

It would be important that authors discuss about the differences between the models for nulliparous and multiparous and their implications. From their estimated coefficients, it seems like they only differ in the role of hypertensive disorders of pregnancy and (obviously) previous vaginal delivery.

The model is of limited utility for settings where there is no information on CES-D. Since most likely this would be the case for most situations, I suggest to run separate models without this variable.

How these results compare to other prediction model? I would suggest to comment about the WHO predictions models, such as C-Model by Souza et al.

Reviewers' comments:

Reviewer's Responses to Questions

**Comments to the Author**

1. Is the manuscript technically sound, and do the data support the conclusions?

Reviewer #1: Partly

Reviewer #2: Partly

2. Has the statistical analysis been performed appropriately and rigorously? 

Reviewer #1: Yes

Reviewer #2: Yes

3. Have the authors made all data underlying the findings in their manuscript fully available?

Reviewer #1: Yes

Reviewer #2: Yes

4. Is the manuscript presented in an intelligible fashion and written in standard English?

Reviewer #1: Yes

Reviewer #2: No

5. Review Comments to the Author

Reviewer #1: The manuscript is relevant and reaches conclusions according to the objectives, but there is a lack of attemption to the analgesic modalities during labour and about the epidural rates. I think it is a very important issue when you are affording CS rates. Statistical analysis has been correctly performed and statistical material has been provided properly. The manuscript is written in correct english, as far as I know (I'm not native english speaker).

The authors should include details about analgesia during labour, epidural rate, and motor block, or at least to detail the epidural analgesic protocol that has been used in this cohort of women.

Reviewer #2: This is an interesting study.

However,

1. Some grammatical etc. errors should be corrected. Examples:

Line 110/Please change “earlypregnancy” to “early pregnancy”

Line 119/Please change “cephalopevlic” to cephalopelvic” (using the advantage of automatic corrections).

2. In the “conclusion”, a real conclusion is expected (and not a detailed iteration of the title).

3. To take advantage of the practical benefits of similar studies, it should be clarified:

a) the indications of emergency CS (ECS)

b) the necessary steps to avoid situations leading to ECS

4. A working calculator could convince that the whole effort is a useful tool to justify or avoid ECS.

6. PLOS authors have the option to publish the peer review history of their article (what does this mean?). If published, this will include your full peer review and any attached files.

Reviewer #1: No

Reviewer #2: No

---

## [Author Response · Author response to Decision Letter 0]

30 Sep 2021

We have provided a file in the response documenting our response to the editor and reviewer comments.

---

## [Decision Letter · Decision Letter 1]

8 Feb 2022

PONE-D-21-10805R1

Prediction of odds for emergency cesarean section: a secondary analysis of the CHILD term birth cohort study

PLOS ONE

Dear Dr. Mandhane,

Thank you for submitting your manuscript to PLOS ONE. After careful consideration, we feel that it has merit but does not fully meet PLOS ONE’s publication criteria as it currently stands. Therefore, we invite you to submit a revised version of the manuscript that addresses the points raised during the review process.

We look forward to receiving your revised manuscript.

Kind regards,

Eduardo Ortiz-Panozo, MD; MSc

Academic Editor

PLOS ONE

Journal Requirements:

Reviewers' comments:

Reviewer's Responses to Questions

**Comments to the Author**

1. If the authors have adequately addressed your comments raised in a previous round of review and you feel that this manuscript is now acceptable for publication, you may indicate that here to bypass the “Comments to the Author” section, enter your conflict of interest statement in the “Confidential to Editor” section, and submit your "Accept" recommendation.

Reviewer #3: (No Response)

Reviewer #4: (No Response)

2. Is the manuscript technically sound, and do the data support the conclusions?

Reviewer #3: Partly

Reviewer #4: Yes

3. Has the statistical analysis been performed appropriately and rigorously? 

Reviewer #3: Yes

Reviewer #4: Yes

4. Have the authors made all data underlying the findings in their manuscript fully available?

Reviewer #3: No

Reviewer #4: Yes

5. Is the manuscript presented in an intelligible fashion and written in standard English?

Reviewer #3: Yes

Reviewer #4: Yes

6. Review Comments to the Author

Reviewer #3: Overall, this study was interesting as they had a large number of samples and a combination of obstetrics and non-obstetrics factors counted for emergency cs prediction tools. However, there are a few things that could be added to improve the quality of the study.

In general, it was suggested to use appropriate written English language. Many paragraphs were considered not effective as repetitive words and redundant sentences were found.

- Abstract:

Keywords: antenatal depression?

Higher AUC in validation set?

Conclusion: Maybe you could paraphrase?

- Introduction

No reference 1? The reference started with number 2

Line 64-66, 66-68: please paraphrase

- Methods & Results

When dividing the data? After recruitment or after applying exclusion criteria? Please specify with number too

The division of training and validation set (80% and 20%) was based on what?

The use of CESD in methods, please put a reference

Diagnosis criteria for hypertensive disorder and diabetes mellitus complicating pregnancy? please put reference too

The Adjusted OR, please specify, adjusted with what?

- Analysis:

This study focuses in comparing vaginal birth and emergency cs. Since you put scheduled cs in the table 1, why don’t you also compare the scheduled cs and the emergency cs? The findings might be interesting and can be added in the discussion too.

- Discussion:

Please add discussion to compare other studies for emergency cs scoring systems. Not only study for cs prediction in general. Such as a current previous study which develop a scoring system for emergency cs by maternal-fetal perinatal characteristics. Although that one was different from yours, maybe you can add this in the discussion.

Typo: Line 144, 155, 249

- Reference:

A number of references were considered too old, there are many newer references that could be used related to the issue

- Supplementary materials:

Table 1: acupressure? Reference?

Table 2 in supplementary files similar to Table 1 in the manuscript? No need to mention it in supplementary files.

Reviewer #4: I have participated in the review process only in this final round. My feeling is that authors have addressed the comments of the reviewers, although I would defer to them in the decision on to what extent it has been achieved. My only suggestion would be the following: the authors have done a good job attending the reviews, especially the one related to the comparison with other tools predicting the risk of C-Section. However, I think it is still possible in the discussion to stress what are the advantages of this tool. The authors must make a strong case for why this new tool is necessary, and the advantages of using this one instead of other existing tools up front.

7. PLOS authors have the option to publish the peer review history of their article (what does this mean?). If published, this will include your full peer review and any attached files.

Reviewer #3: **Yes: **Rima Irwinda

Reviewer #4: **Yes: **BERNARDO HERNANDEZ PRADO

---

## [Author Response · Author response to Decision Letter 1]

21 Mar 2022

Review Comments to the Author

Reviewer #3: Overall, this study was interesting as they had a large number of samples and a combination of obstetrics and non-obstetrics factors counted for emergency cs prediction tools. However, there are a few things that could be added to improve the quality of the study.

In general, it was suggested to use appropriate written English language. Many paragraphs were considered not effective as repetitive words and redundant sentences were found.

- Abstract:

Keywords: antenatal depression?

Higher AUC in validation set?

Conclusion: Maybe you could paraphrase?

Response: Thank you for the suggestion. We have updated the abstract accordingly. Antenatal depression is synonymous with perinatal depression.

- Introduction

No reference 1? The reference started with number 2

Response: Updated the reference. 

Line 64-66, 66-68: please paraphrase

Response: Rephrased the sentences (line 60-67)

Indications for scheduled CS can be divided into absolute and relative indications (1). Absolute indications for scheduled CS include cephalopelvic disproportion, placenta previa, abnormal lie and presentation. Prior CS delivery is a relative indication for scheduled CS ((1–5) . 

Emergency CS is indicated when acute obstetrical complications that threaten the life of the mother and/ or the fetus including fetal distress and antepartum hemorrhage develop. Intrapartum factors such as labour dystocia, fetal distress, and umbilical cord prolapse are absolute indications for emergency CS (1,6,7)

- Methods & Results

When dividing the data? After recruitment or after applying exclusion criteria? 

Response: The data were divided after applying the exclusion criteria. 

Please specify with number too. The division of training and validation set (80% and 20%) was based on what?

Response: The division of training and validation data set (80% vs. 20%) was based on random splitting. Updated accordingly (line 131-134).

“Data analysis steps to develop an emergency CS score are described in Figure 1. A total of 2,836 pregnant women met the inclusion criteria. First, the data were randomly divided into two groups: a training dataset (80% of the sample, n=2,269) and a validation dataset (20% of the sample, n=567).”

The use of CESD in methods, please put a reference. 

Response: Reference added (line 108).

Diagnosis criteria for hypertensive disorder and diabetes mellitus complicating pregnancy? please put reference too

Response: References added (line 106)

The Adjusted OR, please specify, adjusted with what?

Response: updated accordingly in the methods section (line 152-153)

“The final model was adjusted for maternal height, BMI, CESD-score, hypertensive disorders of pregnancy, history of previous vaginal delivery and hospital CS rate.”

- Analysis:

This study focuses in comparing vaginal birth and emergency cs. Since you put scheduled cs in the table 1, why don’t you also compare the scheduled cs and the emergency cs? The findings might be interesting and can be added in the discussion too.

Response: Thank you for the suggestion. The main purpose of this study is to develop an emergency CS risk prediction tool. The study excluded major indications for scheduled CS (i.e. cephalopelvic disproportion, placenta previa, previous CS delivery) in the analysis. Additionally, several tools for predicting CS already exist.

- Discussion:

Please add discussion to compare other studies for emergency cs scoring systems. Not only study for cs prediction in general. Such as a current previous study which develop a scoring system for emergency cs by maternal-fetal perinatal characteristics. Although that one was different from yours, maybe you can add this in the discussion.

Response: We updated with the emergency CS scoring system (line 253-254).

“The emergency CS risk scoring system by Guan et al (2020) (3), with an AUC of 0.79, included intrapartum factors such as quantity and color of the amniotic fluid.”

Typo: Line 144, 155, 249

Response: Typo errors corrected (lines 145, 158, 252)

- Reference:

A number of references were considered too old, there are many newer references that could be used related to the issue

Response: Updated the references

- Supplementary materials:

Table 1: acupressure? Reference?

Response: reference updated

Table 2 in supplementary files similar to Table 1 in the manuscript? No need to mention it in supplementary files.

Response: Table 1 in the manuscript shows the demographic, antenatal and obstetric characteristics associated with mode of delivery. Table 2 in supplementary files presents the demographic, antenatal and obstetric characteristics of Training and Validation data (line 134-135)

“The demographic, antenatal and obstetric characteristics of training and validation data set was shown in S2 Table. “

Reviewer #4: The authors have done a good job attending the reviews, especially the one related to the comparison with other tools predicting the risk of C-Section. However, I think it is still possible in the discussion to stress what are the advantages of this tool. The authors must make a strong case for why this new tool is necessary, and the advantages of using this one instead of other existing tools up front.

Response: We have updated accordingly. (line 274-277)

“The components of our eCS score including maternal age, height, BMI, childbirth history and hypertensive disorders of pregnancy are often collected as part of routine prenatal care. Additionally, unlike prior emergency CS tools, our score does not include any intrapartum data allowing for application at any point during pregnancy.”

---

## [Editor Report · Decision Letter 2]

31 Mar 2022

PONE-D-21-10805R2Prediction of odds for emergency cesarean section: a secondary analysis of the CHILD term birth cohort studyPLOS ONE

Dear Dr. Mandhane,

Thank you for submitting your manuscript to PLOS ONE. After careful consideration, we feel that it has merit but does not fully meet PLOS ONE’s publication criteria as it currently stands. Therefore, we invite you to submit a revised version of the manuscript that addresses the points raised during the review process.

 I apologize for the length of time that it is taking to process your manuscript. The COVID-19 pandemic has slowed our process considerably by affecting availability and response of reviewers. See Additional Editor Comments below.

We look forward to receiving your revised manuscript.

Kind regards,

Eduardo Ortiz-Panozo, MD; MSc

Academic Editor

PLOS ONE

Journal Requirements:

Additional Editor Comments (if provided):

In my view, authors have addressed all reviewers' comments. Lastly, I would request authors to comment on the implications of their tool's low sensitivity in the discussion. Does that mean the tool works better as a "rule-in" rather than a "rule-out" test? And if so, do the relatively high AUC and accuracy result not from identifying (predicting) those who need an emergency CS but those who would not need one?

Please also confirm that PPV in the validation sample is 63% (Table 2).
---

## [Author Response · Author response to Decision Letter 2]

20 Apr 2022

Thank you for the insightful comments and the opportunity to further improve our manuscript. We have provided responses to each of the editor’s comments and have amended the manuscript accordingly. 

Query 1: I would request authors to comment on the implications of their tool's low sensitivity in the discussion. Does that mean the tool works better as a "rule-in" rather than a "rule-out" test? And if so, do the relatively high AUC and accuracy result not from identifying (predicting) those who need an emergency CS but those who would not need one?

Response: We agree that additional clarification of the tool will help the readers understand the tool’s applicability in practice. The high specificity and low sensitivity suggest that the tool is better at determining who will not need an eCS (a rule-out test). The editor is correct that the high AUC and accuracy are from identifying those who do not need an eCS (the majority of women). We would recommend that women who screen positive have closer pre-natal follow-up rather than scheduling a CS. The tool also identifies areas for potential intervention to lower an individuals risk for an eCS. We have added the following to the discussion (Lines 305 – 311)

 “The high specificity and low sensitivity suggest that the tool is good at determining who will not need an eCS (a rule-out test). As such, we would recommend that women who screen positive have closer pre-natal follow-up. The tool identifies areas for potential intervention to lower an individual’s risk for an eCS.”

Query 2: Please also confirm that PPV in the validation sample is 63% (Table 2).

Response: We have confirmed that the PPV of the validation sample is 63%. We have attached the classification table for the validation dataset in this response (appendix 1 of the response).

---

## [Editor Report · Decision Letter 3]

26 Apr 2022

Prediction of odds for emergency cesarean section: a secondary analysis of the CHILD term birth cohort study

PONE-D-21-10805R3

Dear Dr. Mandhane,

We’re pleased to inform you that your manuscript has been judged scientifically suitable for publication and will be formally accepted for publication once it meets all outstanding technical requirements.

Kind regards,

Eduardo Ortiz-Panozo, MD; MSc

Academic Editor

PLOS ONE
---

## [Editor Report · Acceptance letter]

4 May 2022

PONE-D-21-10805R3 

Prediction of odds for emergency cesarean section: a secondary analysis of the CHILD term birth cohort study 

Dear Dr. Mandhane:

I'm pleased to inform you that your manuscript has been deemed suitable for publication in PLOS ONE. Congratulations! Your manuscript is now with our production department. 

Kind regards, 

on behalf of

Dr. Eduardo Ortiz-Panozo 

Academic Editor

PLOS ONE